# Comparison of Motor Difficulties Measured in the First Year of School among Children Who Attended Rural Outdoor or Urban Conventional Kindergartens

**DOI:** 10.3390/ijerph192114158

**Published:** 2022-10-29

**Authors:** Ina Olmer Specht, Sofus Christian Larsen, Jeanett Friis Rohde, Jane Nautrup Østergaard, Berit Lilienthal Heitmann

**Affiliations:** 1Research Unit for Dietary Studies, The Parker Institute, Bispebjerg and Frederiksberg Hospital, The Capital Region, 2000 Frederiksberg, Denmark; 2Steno Diabetes Center Aarhus, Aarhus University Hospital, Central Denmark Region, 8200 Aarhus, Denmark; 3The Boden Group, Faculty of Medicine and Health, Sydney University, Sydney, NSW 2006, Australia; 4Section for General Practice, Department of Public Health, University of Copenhagen, 1165 Copenhagen, Denmark

**Keywords:** day care, nature, selection bias, outdoor life, childhood, pre-school

## Abstract

Background: Kindergartens can potentially contribute substantially to the daily level of physical activity and development of motor skills and might be an ideal setting for improving these as a public health initiative. We aimed to examine whether children from rural outdoor kindergartens had a lower risk of motor difficulties than children from urban conventional kindergartens. Methods: Motor test results were measured during the first school year by school health nurses using a six-item test of gross- and fine motor skills (jumping, handle a writing tool, cutting with a scissor following a line, one-leg stand on each leg, throwing and grabbing). Register-based information was available on potential confounding factors. Results: We included 901 children from outdoor kindergartens and 993 from conventional kindergartens with a mean (SD) age of 6.5 years (0.4). The children from the two types of kindergarten differed according to demographic information, with outdoor kindergarten children more often being from more affluent families (long maternal education level: 47.5% vs. 31.0%, *p* < 0.0001) and fewer girls attending the outdoor kindergartens (42.7% vs. 49.5%, *p* = 0.003). In the adjusted models, we found no evidence of differences in the risk of motor difficulties between children attending either type of kindergarten (OR: 0.95, 95%CI: 0.71; 1.27, *p* = 0.72). Conclusion: Our results do not support outdoor kindergartens as a potential intervention to improve motor abilities among children. Randomized controlled trials are needed to confirm these findings.

## 1. Introduction

A sedentary lifestyle with much time spent indoor and/or in front of a screen has become more and more the norm, even among children as young as 3–5 years old, especially among those from urban areas [1]. Studies have shown that a high amount of time spent in front of screens is associated with low levels of physical activity and with an unhealthy lifestyle, i.e., unhealthy diet, obesity, depressive symptoms, low self-esteem, and poor motor skills [2,3,4]. Conversely, studies support that physical and outdoor activities among young children promotes physical and mental health as well as motor skills [5,6,7]. Higher physical activity and better motor skills have further been shown to associate with better cognitive development [8,9,10].

Many pre-school children in Denmark attend up to 40 h of kindergarten per week [11], and since up to 97% of young children attend kindergartens in the Scandinavian counties including Denmark [12] and 87% in the other OECD counties [13], kindergartens are institutions that contribute substantially to the daily level of physical activity. They can thus be an ideal setting for public health initiatives to find ways to improve physical activity habits and active outdoor lifestyles among children.

In Scandinavian countries, there are overall two types of kindergartens with the same number of care personnel per child: (1) the conventional kindergarten, where children spend time both indoor playing with toys, drawing and tumbling, and outdoor in the kindergarten playground, and (2) the outdoor kindergartens, a Danish invention from 1952, where almost all hours during the day are spent outdoor with free or planned play activities, usually in forests or in rural areas without a formal playground [14,15,16]. Children attending outdoor kindergartens are often living in urban areas, and thus are transported by bus from the city to the outdoor kindergarten setting.

Positive opinions in relation to health, physical activity and motor development of children from outdoor kindergartens are commonly reported by the Danish media [17]. The evidence for these health benefits is, however, limited to only a few small Scandinavian studies. For instance, a small Danish study investigated motor development, level of awareness, ingenuity and frequency of sickness absence among children attending one outdoor kindergarten compared to one conventional kindergarten [18]. In that study, the children from the two types of kindergartens were examined at baseline and after 10 months. Results showed that children from the outdoor kindergarten scored better on all outcomes examined. Similarly, another small study among 5–7 year old kindergarten children from Norway investigated versatile play in an outdoor forest environment (*n* = 46 children) compared to a kindergarten playground (*n* = 29 children) [19]. That study showed that over a period of 9 months, the children who daily played 1–2 h in the forest gradually improved their motor ability more than the children who spent 1–2 h daily at the kindergarten playground [19]. Moreover, in a previous study, we showed that during kindergarten time children were more physically active (more steps/hour) in the outdoor kindergartens than in the conventional kindergartens [20]. While only few previous studies have investigated health outcomes differences related to outdoor and conventional kindergartens several studies have investigated relations between nature contact and child health. Indeed, a systematic review including 296 studies evaluated the evidence for relationships between nature contact and children’s health and reported finding direct associations between nature and physical activity in about 2/3 of all studies [21]. However, the majority of the included studies were cross-sectional and examined physical activity in relation to residential green spaces by land-use data or Normalized Difference Vegetation Indexes [21]. It has also been proposed that access to green schoolyards, as opposed to asphalt-based schoolyards, improved physical activity and socioemotional health among the children [22]. The uneven ground, and the more room for ingenuity have been proposed as the main factors explaining the higher physical activity level and motor skills observed among children playing in outdoor green environments compared to asphalt covered or indoor environments [19,23].

Common to the majority of the previous studies that examined potential health benefits of outdoor kindergartens as well as nature per se are, that in addition to having examined small samples of children only, many of them also had a high risk of selection or confounding bias due to lack of comparability among investigated groups, especially in relation to socio economic status (SES) [21]. In this regard, we previously showed in a large register-based study that children attending conventional kindergartens, compared to children who attended outdoor kindergartens, differed substantially according to parental socio- and early childhood demographics [24]. Outdoor kindergarten children came from higher socio demographic families, which can cause selection bias and it is thus highly relevant to consider socio demographics when investigating health outcomes related to kindergarten type attainment [24]. Well-designed studies are thus needed to confirm whether outdoor kindergartens in fact are promoting health including improving motor abilities.

In Denmark, all children attend a health nurse examination during the first year of school, and in most municipalities a motor ability test is included. These measures combined with the register-based information on parental socio- and early childhood demographics, gives us a unique opportunity to examine whether outdoor kindergartens could be a potential early preventive strategy towards better motor skills.

Our primary objective for this study was thus to examine whether children from outdoor kindergartens had a lower risk of motor difficulties compared to children from conventional kindergartens.

We hypothesized that compared to children attending conventional kindergartens, children attending outdoor kindergartens have lower risk of motor difficulties, due to a higher level of physical activity in the outdoor kindergarten.

## 2. Materials and Methods

The present study was part of the ‘Outdoor kindergartens—the healthier choice?’ (ODIN) study which was initiated with the main goal to investigate pedagogic and didactic practice and child health among children attending outdoor kindergartens compared to children attending conventional kindergartens. In the ODIN study we had data from a total of 5077 children, of which 2434 attended outdoor kindergartens and 2643 attended conventional kindergartens, based on data obtained during the period 2014–2019 from the Copenhagen Municipality and the period 2011–2019 from the Aarhus Municipality, the two largest cities of Denmark. Both municipalities provided civil registration numbers (CPR-numbers) and health information gathered by health nurses during the first year of the child’s life, and from the school health surveys. The present study included all children who went to an outdoor kindergarten in the above referred periods and a random subsample of all children from conventional kindergartens, from the same areas of parental residence. These data were merged with data from the Danish registers to gather baseline demographic and health information on parents and children. In the present study, motor development was only available from the Copenhagen Municipality, as Aarhus Municipality did not test for motor development. Of the 2093 children from Copenhagen Municipality who had information on motor skills recorded after school entry, we further excluded 199 with missing information on covariates, ending up with a final sample size of 1894 children.

We further had information on motor ability results taken by the health nurse form 227 children at age 8–10 months. This test includes 13 sub-test investigating the child’s interaction, attention, and reaction to certain visual and sound impressions. The test is designed to detect children with contact disorders, hearing and visual impairments, unsatisfactory environmental stimulation and stunted development.

### 2.1. Exposure Assessment

The exposure was attending either an outdoor kindergarten or a conventional kindergarten. The children had to have attended either an outdoor kindergarten or a conventional kindergarten in the years they lived in the municipality, thus the exposure time for the individual child differed according to how many years the child had lived in the municipality. Type of kindergarten and when the child was enrolled or ended kindergarten were registered by the municipalities.

### 2.2. Outcome Assessment and Covariates

The outcome was the result of the first mandatory motor test conducted as part of the health examination carried out in the first year of school, approximately at age 6 years. The motor test included six sub-tests; ‘dynamic jump on the spot’, ‘handle a writing tool’, ‘cutting with a scissor following a line’, ‘One-leg stand for 10 s on each leg’, ‘throwing’ and ‘grabbing’ a small bag filled with rice. In the throwing and grabbing tests, 7 out of 10 should be successful to pass the test. Each test can obtain one remark if not managed by the child. In the present study, motor difficulties were defined based on having two or more remarks from the total of the six sub-tests. This number of remarks was chosen based on results from a Danish report investigating motor development among 5963 children from 13 Danish municipalities (not including Copenhagen and Aarhus) during the first year of school in the school year 2018/19 [25]. The municipalities included in the report had nine motor tests whereas the Copenhagen Municipality have six tests, only. In the report, having three or more remarks were considered as motor difficulties based on the nine motor tests performed, thus we defined a child as having motor difficulties when having two or more remarks among the six available tests. Since one remark also can have consequences for the child’s everyday life [25], we also investigated this in sensitivity analysis.

To gather information on covariates, the available health nurse data including the motor test results, and the child weight and height were sent to Statistics Denmark where merging using the CPR-number with information from The Danish Medical Birth Register and population data from Statistic Denmark took place.

Potential confounders/covariates were preterm birth (born before gestational week 37, yes/no), maternal education (Basic [basic school 8th–10th class]; Short [general upper–secondary education, short-cycle higher education or vocational education and training]; Medium [medium-cycle higher education or bachelor]; and Long [long-cycle education and PhD]), maternal origin (Western/non-Western), sex of the child (female/male), birth weight (continuous, g), years spend in kindergarten (continuous), and age at motor test (continuous, years).

### 2.3. Statistical Analysis

A detailed statistical analysis plan, including estimates of statistical power and precision, was approved by all authors prior to conducting any statistical analyses (Appendix A).

In the primary analysis, logistic regression was used to examine the odds ratio (OR) of motor difficulties with two or more remarks (yes/no) among children in outdoor compared to conventional kindergartens, based on the first recorded motor test taken during the first school year. Results are presented from a crude analysis with information on motor development and kindergarten type only, and a fully adjusted model with added information on all additional covariates described above.

In sensitivity analysis, we investigated motor difficulties related to having one or more remarks (yes/no), only, both in crude and fully adjusted analyses.

We further tested for effect modification of several pre-defined co-variates which have been suggested earlier as primary predictors of motor development. Effect modification by sex, maternal education level (low vs. high) and the total time in years spend in outdoor or conventional kindergartens (continuous variable) were investigated in separate analysis by adding each variable and a product term to the fully adjusted model. Subgroup analyses specifically for children of low educated mothers (defined as mothers with a basic or short education) and high educated mothers (defined as mothers with a medium or long education) were also conducted.

Finally, as crude ad hoc tests, we investigated if motor abilities (one or more remarks out of 13 motor sub-tests) tested by health nurses during the first year of life (*n* = 227) were associated with motor abilities (two or more remarks) at school age using logistic regression, and further if children with missing covariates and thus deleted from the analysis differed according to motor test abilities.

All statistical tests were two-sided with a significance level at 0.05 and analyses were performed using SAS Enterprise Guide 8.3 on a secure platform at Statistics Denmark.

### 2.4. Ethical Considerations

Permission from the municipalities to send information to Statistics Denmark was granted. Permission from the Ethical Committee was evaluated not to be relevant (journal nr.: H-19053587). Permissions from the Capital Region Data Agency and the Danish Patient Safety Authority were granted (Journal nr.: P-2020-54 and 31-1521-8, respectively). Registries relevant for the study are accessible and the collection of data from these registries is in accordance with accepted ethical principles for informed consent according to the Declaration of Helsinki.

## 3. Results

A total of 901 children from outdoor kindergartens and 993 children from conventional kindergartens were included in the analyses. Characteristics of the children divided by type of kindergarten are shown in Table 1. The majorities of the investigated characteristics differed according to kindergarten type with children attending outdoor kindergartens more often than children attending conventional kindergartens having mothers or fathers with longer educations (Long: 47.5% vs. 31.0%, *p* < 0.0001 and 43.2% vs. 25.9%, *p* < 0.0001, respectively), and with Western origin (95.8% vs. 72.5%, *p* < 0.0001, and 94.2% vs. 73.3%, *p* < 0.0001, respectively); also fewer children attending outdoor kindergartens were born preterm 3.8% vs. 6.6%, *p* = 0.007) and had a female sex (42.7% vs. 49.5%, *p* = 0.003). BMI z scores were lower at school entrance among children from outdoor kindergarten as compared to conventional kindergarten children (mean (SD): 0.07 (0.98) vs. 0.20 (1.09), *p* = 0.005), whereas no statistically significant differences were seen for age at kindergarten enrolment (*p* = 0.36), time spent in kindergarten (*p* = 0.94), and age at motor test (6.5 years (0.4), *p* = 0.27).

The crude model investigating motor skills at school entrance for children with ≥2 remarks from the motor test compared to <2 remarks, showed a non-statistically significantly lower odds of a poorer motor skills test results among children from outdoor kindergartens as compared to children from conventional kindergartens (OR crude: 0.79, 95%CI: 0.61; 1.03, *p* = 0.08, Table 2). The same tendency was shown in the crude analysis when children were categorized according to ≥1 or no remarks (OR crude: 0.84, 95%CI: 0.69; 1.01, *p* = 0.07, Table 2). However, in the fully adjusted models, no evidence of differences in motor skills were shown between children attending outdoor and conventional kindergartens (≥2 remarks OR adj.: 0.95, 95%CI: 0.71; 1.27, *p* = 0.72 and ≥1 remarks OR adj.: 0.93, 95%CI: 0.75; 1.15, *p* = 0.51, Table 2).

No statistically significant effect modification between kindergarten type and sex, maternal education or years spent in kindergarten were shown (sex: *p* = 0.54; maternal education: *p* = 0.07, and years in kindergarten: *p* = 0.74). However, since a borderline significant effect modification was shown for maternal education, we stratified the analyses by education (low vs. high). No significant between group differences in motor skills in subgroups with mothers of low or high education level were shown (outdoor versus conventional kindergartens: low maternal education OR = 0.67, 95%CI = 0.39; 1.15, *p* = 0.15; and high maternal education OR = 1.08, 95%CI = 0.76; 1.54, *p* = 0.66, (Figure 1).

Finally, no association with motor test results at school entrance was shown among the 227 children with motor test results taken by the health nurse during the first year of life (OR = 1.32, 95%CI = 0.58; 2.99, *p* = 0.51). Additionally, children without missing covariates (*n* = 1994) did not differ according to motor test result compared to children with missing covariates (*n* = 199, *p* = 0.62).

## 4. Discussion

Considerable socio-demographic differences were shown between children from outdoor kindergartens as compared to children from conventional kindergartens, with a higher proportion of children from less affluent families in conventional kindergartens. Children from less affluent families have been shown to be less physically active and display worse motor abilities as compared to children from more affluent families [26]. In the crude analysis, there was a statistically non-significant tendency for lower risk of motor difficulties taken by the school health nurse during the first year of school among the children from outdoor compared to children from conventional kindergartens, however after adjusting for potential confounders like SES, no differences were seen between the children from the two types of kindergartens. Significant associations between outdoor kindergartens and motor abilities have, however, been shown previously [18,19,27].

Since previous studies show that, as compared to children from more affluent families, children from less affluent homes are generally more sedentary in their spare time [28], spend less time outdoor and use nature less [29], children from these families could be expected to benefit most from outdoor kindergartens. However, in our stratified analysis by level of maternal education we did not see between group differences, although the odds ratio pointed towards a lower risk of motor difficulties among children from outdoor compared to conventional kindergartens for those children with less educated mothers. Controversially, a German cluster randomized intervention study promoting more physical activity in the intervention kindergartens managed by teachers over a period of one year, showed a significant improvement in endurance performance; however, when stratified by SES this was only apparent among children from high or medium SES families, and not among children from low SES families [30]. These results point towards that preventive interventions in kindergartens should differ according to socioeconomic status of the child or a more intense intervention is needed among less educated families to gain most effective results.

More boys than girls attended outdoor kindergartens. Indeed, in our previous study we showed that parents to children from outdoor kindergartens often valued physical activity higher than parents choosing conventional kindergartens, and they choose to a higher degree an outdoor kindergarten, if they perceived their child as active [24]. In accordance, a previous study showed that boys seem to generally be more active during day care hours than girls [31] and boys might thus also be more active before kindergarten enrolment. We did, however, not find any effect modification by sex in this study.

The study has some potential limitations. We did not show a significant difference in motor skills among children from the two kindergarten types although the odds ratio among outdoor kindergarten children as compared to conventional kindergarten children was below 1. In our pre-study power calculation, we showed that we had a power of 80% to detect odds ratios < 0.6 or >1.5. Thus, with a larger sample size we might have been able to show a significant association. Unfortunately, we were not able to include more children, since the municipalities did not store information on which type of kindergarten children attended for a longer period than approximately four years. Another limitation may be the lack of objectively measured physical activity level among the children from the two kindergarten types. Thus, we do not know if the children from outdoor kindergartens in fact were more physically active than children from conventional kindergartens. However, in our previous post-test–pre-test study among children from rotating kindergartens, which are kindergarten where children alternate between spending one week in the outdoor kindergarten setting and the next in the conventional kindergarten setting, we showed that children were more physically active, as measured by accelerometer, during kindergarten hours when in the outdoor environment [20]. A difference that has also been shown in a previous qualitative study [32]. However, we also found a compensation in activity during non-kindergarten time which resulted in overall no significant difference between daily physical activity for children from the two kindergarten types [20]. Another potential limitation, which also might explain the overall lack of significant differences in daily physical activity as showed in the previous study [20], is the lack of information about sedentary transportation time. Children in outdoor kindergartens are transported by bus to the outdoor kindergarten and might sit in the bus for more than an hour daily which potentially could affect their motor development. 

We did not have access to measured motor skills before kindergarten enrolment for all children; however, in our ad hoc analysis, among the 227 children with a health nurse motor test performed during the first year of life and a motor test at school entry, we did not show an association, but the sample size was small. The school motor test used by the health nurses are not validated or made for research purpose, and only includes six sub-tests of fine- and gross motor abilities. The motor test is used as a public health initiative for further evaluation by a physiotherapist or specialist in motor development if needed. However, it is a strength that the motor test was carried out by trained school health nurses. These tests are mandatory in the Copenhagen Municipality and performed during school hours, thus we expect a minimum of selection bias. Further, the children with missing covariates, and thus not included in the analysis, did not differ according to motor skills from those included.

Another strength to this study was the register-based design with information on potential confounding factors, although residual confounding can always occur. The majority of previous studies that investigated the relationship between nature exposure or outdoor kindergartens and physical activity or motor skills among children did not adjust for confounders, such as parental SES and the access to, actual use, or quality of the nature exposure, which might explain the reported direct associations observed in these studies [21,27]. Although we did not investigate the exact time spent outside in the nature in the two kindergarten environments, it is well-known that the structural fundament of the outdoor kindergartens as well as the didactive pedagogic practice are based on play in the nature and outdoor life [16].

Still, outdoor kindergartens might be effective in countries with more pronounced socioeconomic differences that in the Scandinavian countries, but our results may not be generalizable to children from other countries.

## 5. Conclusions

Although nature environments have been shown to promote physical activity as compared to indoor environments, we did not find differences in the risk of motor difficulties among children from outdoor versus conventional kindergartens. Outdoor kindergartens have been promoted as beneficial for motor skills development and health compared to conventional kindergartens, but selection bias related to socio economy, with more children from affluent families attending outdoor kindergartens, may explain previous positive findings. Future studies must be aware of selection bias in kindergarten-type attendance, and thus randomized controlled trials are preferred.

## Figures and Tables

**Figure 1 ijerph-19-14158-f001:**
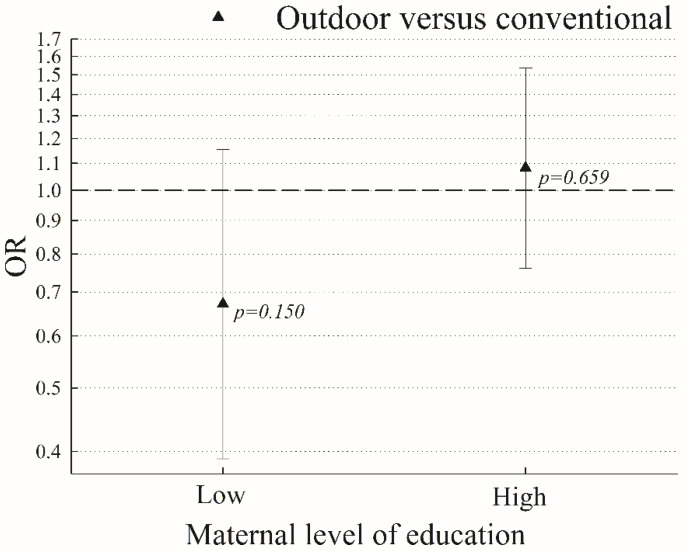
Adjusted odds ratios (OR) for motor skills test results among children from outdoor versus conventional kindergartens stratified by maternal education level.

**Table 1 ijerph-19-14158-t001:** Participant characteristics of children in outdoor kindergartens versus conventional kindergartens. Results are presented as percentage or mean and standard deviation (SD).

	Outdoor Kindergarten *n* = 901			Conventional Kindergarten *n* = 993			
	*n*	%	Mean (SD)	*n*	%	Mean (SD)	*p*
Female sex	385	42.7		491	49.5		0.003
Age at kindergarten enrolment, years	808		3.7 (0.8)	893		3.6 (0.8)	0.36
Time spent in kindergarten, days	901		757 (372)	993		760 (373)	0.94
Age at first school motor measurement, years	901		6.5 (0.4)	993		6.5 (0.4)	0.27
BMI z score at motor test	891		0.07 (0.98)	989		0.20 (1.09)	0.005
Birth weight, g	901		3503 (513)	993		3433 (540)	0.11
Preterm birth	34	3.8		65	6.6		0.007
Maternal education							<0.0001
Basic	47	5.2		186	18.7		
Short	148	16.3		258	26.0		
Medium	278	30.9		241	24.3		
Long	428	47.5		308	31.0		
Paternal education							<0.0001
Basic	63	7.3		165	17.6		
Short	250	28.8		345	36.8		
Medium	181	20.8		184	19.6		
Long	375	43.2		243	25.9		
Maternal country of origin							<0.0001
Western	863	95.8		720	72.5		
Non-Western	38	4.2		273	27.5		
Paternal country of origin							<0.0001
Western	827	94.2		709	73.3		
Non-Western	51	5.8		258	26.7		

**Table 2 ijerph-19-14158-t002:** Motor skills during the first school year among children from outdoor versus conventional kindergartens.

	OR	95%CI	*p*
Motor, ≥2 remarks			
Crude model 1	0.79	0.61; 1.03	0.08
Adj. model 1 ^1^	0.95	0.71; 1.27	0.72
Motor, ≥1 remark			
Crude model 2	0.84	0.69; 1.01	0.07
Adj. model 2 ^1^	0.93	0.75; 1.15	0.51

^1^ Adjusted for sex, birth weight, preterm birth, maternal education, maternal origin, age at motor test.

## Data Availability

Data is available on a secure platform on Statistics Denmark’s homepage. To get access to data, an application has to be sent to The Parker Institute and permission has been granted by the Danish Data Protection Agency, and Statistics Denmark.

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
