# Peer review of "Comparison of Motor Difficulties Measured in the First Year of School among Children Who Attended Rural Outdoor or Urban Conventional Kindergartens"

_ijerph, 2022, doi:10.3390/ijerph192114158_

Round 1
Reviewer 1 Report
Title: Motor difficulties during the first year of school among children who attended rural outdoor or urban conventional kindergartens
Article Type: Article
Summary
In this article, the authors compared the motor difficulties among children from rural outdoor and urban conventional kindergartens. Subjects were 901 children from outdoor kindergartens and 993 from conventional kindergartens who were assessed. The results show no difference between these two groups of children in regards of the risk of motor difficulties.
Evaluation
The topic of this study is interesting for publication in the IJERPH. The sample size and the design for the study is appropriate to answer the research questions, and the paper is well written. However, there are a lot of points should be addressed by the authors, in order to improve the quality of the manuscript.
Points and suggestions
- Please add mean and SD for age for both groups in the abstract.
- Please add the name of motor test and items in the abstract.
- Please speak in detail about the results in the abstract.
- Please add a better conclusion to the abstract not only saying the is a difference between these two groups.
- Please, at the end of the introduction, the research gap and the necessity of doing this research should be mentioned much better, so that the reader can better understand the necessity of doing this research.
- Please add more information such as gender, socioeconomic level of the children and their mean age in the method section.
- To measure children's fundamental motor skills, why didn't you use common standard tests such as MABC-3 and or TGMD-3?
- Line 123, Please add a reference for your used TEST(the health nurse form of motor ability)
- Please add the effect size for all comparison in table 1 and 2.
Author Response
Reviwer 1
Summary
In this article, the authors compared the motor difficulties among children from rural outdoor and urban conventional kindergartens. Subjects were 901 children from outdoor kindergartens and 993 from conventional kindergartens who were assessed. The results show no difference between these two groups of children in regards of the risk of motor difficulties.
Evaluation
The topic of this study is interesting for publication in the IJERPH. The sample size and the design for the study is appropriate to answer the research questions, and the paper is well written. However, there are a lot of points should be addressed by the authors, in order to improve the quality of the manuscript.
Answer: We would start by thanking the reviewer to take the time to review this paper. We hope that our answers are satisfactory.
Points and suggestions
- Please add mean and SD for age for both groups in the abstract.
Answer: Thank you for a relevant comment. Age and SD has now been included.
- Please add the name of motor test and items in the abstract.
Answer: Unfortunately, the motor test used by health nurses in Danish school does not have a name since it is part of a larger health examination performed the first year of school. The motor tests used might also differ according to municipality, however, since we only include one municipality in the study, this is not an issue. It is the same six tests all health nurses in the included municipality use when evaluating the children. The evaluation is not made for research but as a public health initiative to further evaluation if needed. We have included the following in the Abstract and the Discussion:
‘Motor test results were measured during the first school year by school health nurses using a six-item test of gross- and fine motor skills (jumping, handle a writing tool, cutting with a scissor following a line, one-leg stand on each leg, throwing and grabbing).’ Lines 20-22.
In the Discussion section lines 326-330:
‘The school motor test used by the health nurses are not validated or made for research purpose, and only includes six sub-tests of fine- and gross motor abilities. The motor test is used as a public health initiative for further evaluation by a physiotherapist or specialist in motor development if needed. However, it is a strength that the motor test was done by trained school health nurses.’
- Please speak in detail about the results in the abstract.
Answer: Thank you we agree and have included the following:
‘The children from the two types of kindergarten differed according to demographic information among others with outdoor kindergarten children more often being from higher affluency families (long maternal education level: 47.5% vs. 31.0%, p <0.0001), also fewer girls attended the outdoor kindergartens (42.7% vs. 49.5%, p = 0.003).’ Lines 24-28.
And
‘In the adjusted models we found no evidence of a differences in the risk of motor difficulties between children attending either type of kindergarten (OR: 0.95, 95%CI: 0.71;1.27, P = 0.72).’ Line 28.
- Please add a better conclusion to the abstract not only saying the is a difference between these two groups.
Answer: Thank you, we have rewritten the conclusion:
‘Our results do not support outdoor kindergartens as a potential intervention to improve motor abilities among children. Randomized controlled trails are needed to confirm these findings.’ Lines 30-32.
- Please, at the end of the introduction, the research gap and the necessity of doing this research should be mentioned much better, so that the reader can better understand the necessity of doing this research.
Answer: Thank you, we have included the following in lines 100-102:
‘Well-designed studies are thus needed to confirm whether outdoor kindergartens in fact are promoting health including improving motor abilities.’
- Please add more information such as gender, socioeconomic level of the children and their mean age in the method section.
Answer: We have respectfully chosen not to include this information in the method section since it is part of the results.
- To measure children's fundamental motor skills, why didn't you use common standard tests such as MABC-3 and or TGMD-3?
Answer: The data on motor ability was gathered by health nurses as part of the mandatory school health examinations that are performed on all children when starting school. The data we got was thus not collected by researchers as part of this project. It would have been preferable to have validated motor results (as now also mentioned in the Discussion lines 326-330) but this was not possible due to the design of the project.
- Line 123, Please add a reference for your used TEST(the health nurse form of motor ability)
Answer: Unfortunately, no reference in English on the motor test exists. The health nurses are trained in using the tests, which are very simple. However, we have now included an extra sentence about the test in the Methods section line 153-154: ‘In the throwing and grabbing tests 7 out of 10 should be successful to pass the test.‘
- Please add the effect size for all comparison in table 1 and 2.
Answer: Thank you for your suggestion, however we respectfully disagree since table 1 is a descriptive table and thus effects sizes are in our opinion not relevant. Rather, effect estimates are given in table 2. It could be argued also to include adjusted measures of absolute risk and the related risk differences, however since the results are all non-significant, it would not change the conclusion and therefore would, in our opinion, not add additional information.
Reviewer 2 Report
Dear Authors,
The study is interesting and properly prepared. However, the title should be changed. I believe that the "Comparison of motor skills of children who attended rural outdoor or urban conventional kindergartens" is better. The current one is strongly focused on motor difficulties, and this is not the purpose of the publication. It focuses on assessing the implementation of exercises and comparing between groups. I also believe that the limitations of the study should be indicated, such as the lack of an accurate daily measurement of traffic in kindergartens.
Indicate the purpose of the research again and describe it in detail.
The title should indicate the essence of the research and the knowledge gap in this field.
Was the body composition of these children tested? This is an important factor that affects mobility. Anthropometric information should be included in the study.
The study takes into account the education of mothers and the time spent in kindergarten but gives little assessment of the respondents and their activity in these places. These data should be measurable, such as body composition and several steps taken. It was also not reported whether the children had previously performed the exercises in the study. Knowing about such movements or performing them once again influences the quality and time of execution.
Add an explanation in table 2.
The title should also include the range of questions and factors influencing children's activity. Most of them are beyond somatic, such as parents' education and birth weight, which do not directly affect the level of activity in kindergarten and children's agility. The conclusions resulting from the study do not result from the activity and nature of the effort in kindergarten, because it was studied to a limited extent.
It needs to be corrected.
Kind regards,
Reviewer
Author Response
Reviewer 2
The study is interesting and properly prepared. However, the title should be changed. I believe that the "Comparison of motor skills of children who attended rural outdoor or urban conventional kindergartens" is better. The current one is strongly focused on motor difficulties, and this is not the purpose of the publication. It focuses on assessing the implementation of exercises and comparing between groups. I also believe that the limitations of the study should be indicated, such as the lack of an accurate daily measurement of traffic in kindergartens.
Answer: Thank you very much for taking the time to review this manuscript and also for the kind words, and relevant comments.
We have changed the title, however we have chosen to keep ‘difficulties’ and not use ‘skills’ since we are looking at difficulties. We divide the six motor tests into two groups (two or more remarks to the test = motor tests difficulties). We have also chosen to keep the part about when the test was done (the first year of school) to underline that this is not a cross-sectional study.
In relation to “daily measurement of traffic in kindergartens”: Also, thank you for this comment, we agree that transportation time is important to include as a limitation in the Discussion section. Hence, we have now included the following in lines 321-325:
‘Another potential limitation, which also might explain the overall lack of significant differences in daily physical activity as showed in the previous study (20), is the lack of information about sedentary transportation time. Children in outdoor kindergartens are transported by bus to the outdoor kindergarten and might sit in the bus for more than an hour daily which potentially could affect their motor development.’